# Development of Toehold Switches as a Novel Ribodiagnostic Method for West Nile Virus

**DOI:** 10.3390/genes14010237

**Published:** 2023-01-16

**Authors:** Antonis Giakountis, Zoe Stylianidou, Anxhela Zaka, Styliani Pappa, Anna Papa, Christos Hadjichristodoulou, Kostas D. Mathiopoulos

**Affiliations:** 1Department of Biochemistry and Biotechnology, University of Thessaly, Biopolis-Mezourlo, 41500 Larissa, Greece; 2Department of Microbiology, Aristotle University of Thessaloniki, 54124 Thessaloniki, Greece; 3Laboratory of Hygiene and Epidemiology, Faculty of Medicine, University of Thessaly, 41222 Larissa, Greece

**Keywords:** West Nile virus, diagnosis, toehold switches, low-cost surveillance, synthetic biology

## Abstract

West Nile virus (WNV) is an emerging neurotropic RNA virus and a member of the genus *Flavivirus*. Naturally, the virus is maintained in an enzootic cycle involving mosquitoes as vectors and birds that are the principal amplifying virus hosts. In humans, the incubation period for WNV disease ranges from 3 to 14 days, with an estimated 80% of infected persons being asymptomatic, around 19% developing a mild febrile infection and less than 1% developing neuroinvasive disease. Laboratory diagnosis of WNV infection is generally accomplished by cross-reacting serological methods or highly sensitive yet expensive molecular approaches. Therefore, current diagnostic tools hinder widespread surveillance of WNV in birds and mosquitoes that serve as viral reservoirs for infecting secondary hosts, such as humans and equines. We have developed a synthetic biology-based method for sensitive and low-cost detection of WNV. This method relies on toehold riboswitches designed to detect WNV genomic RNA as transcriptional input and process it to GFP fluorescence as translational output. Our methodology offers a non-invasive tool with reduced operating cost and high diagnostic value that can be used for field surveillance of WNV in humans as well as in bird and mosquito populations.

## 1. Introduction

West Nile virus (WNV), first isolated in 1937 from the West Nile province of Uganda [1], is a member of the *Flavivirus* genus (*Flaviviridae* family), with a positive ssRNA genome of ~11,000 nucleotides [2] that shows homology to dengue (DENV) and Zika virus (ZIKV) [3,4]. Migratory birds are the main natural reservoir of WNV [5], from which the virus is transmitted to birds and to dead-end hosts such as humans and equines mainly through infected *Culex* spp and secondarily by *Aedes albopictus* mosquitoes [6]. In mosquitoes, after being picked-up by an infected blood meal, WNV copies its genome in epithelial midgut cells, enters the blood lymph and subsequently spreads to the remaining tissues, particularly to the salivary glands of the mosquito where it multiplies again to infection-sufficient levels [7]. Interestingly, WNV is maintained also in mosquito-eggs [2]. In mammals, WNV is locally copied in Langerhans cells of the skin, from which it enters blood circulation, spreads and multiplies in distant organs, such as the spleen and kidneys and occasionally crosses the blood-brain barrier and infects the brain [8]. Mosquito-mediated transmission of WNV, like other arboviruses (arthropod-borne viruses), depends on hosts that develop high levels of viremia (e.g., birds). Unlike birds, WNV levels rapidly diminish in mammals, like humans and horses, with viremia levels of approximately 10^3^–10^5^ pfu/mL and 10^3^ pfu/mL respectively [9,10]. It is estimated that the minimum viremia level for mosquito-mediated transmission is approximately 10^6^–10^7^ pfu/mL [11]. 

In humans, duration of WNV incubation is estimated at 3–14 days, yet most of the infected individuals (~80%) remain completely asymptomatic [12]. A smaller percentage (~20%), demonstrates mild, flu-like symptoms of the disease, while in approximately 1% of patients the virus successfully infects the central nervous system (CNS), resulting in the development of neuroinvasive disease and even death [13]. Infestation of the CNS is the most serious manifestation of the disease, that can develop into meningitis, encephalitis or acute flaccid paralysis [14,15,16,17,18]. It is estimated that over 6 million were infected by WNV in the US till 2015, of which 24,000 were seriously infected and 2300 died by the disease, with a mortality rate of 9.5% among the severe cases [19]. In general, WNV displays a world-wide distribution and is considered the most widespread arbovirus and the primary cause of viral-mediated encephalitis. 

Currently, assessing the incidence of WNV, including asymptomatic cases, is achieved through seroepidemiological studies [20]. Diagnosis of WNV infection can be achieved with serological or molecular methods in the blood or cerebrospinal fluid (CSF) of patients. Viral RNA detection is challenging due to low viremia. The viruria lasts more than viremia, especially in patients with severe symptoms [21]. Serological methods include ELISA and immunofluorescent assays which can detect WNV IgG or IgM antibodies 4 to 8 days post infection [22]. Cross-reactivity with other flaviviruses such as DENV, ZIKV, yellow fever (YFV), Usutu (USUV), tick-borne encephalitis virus (TBEV) or Japanese encephalitis viruses (JEV) presents the biggest challenge for serology, together with the overall lower sensitivity of immunological-based detection. Thus, serological methods must be confirmed by plaque reduction neutralization test (PRNT) or molecular methods [23]. Molecular methods rely on the detection of the WNV genomic RNA and primarily involve nested-PCR, real time-quantitative PCR (RT-qPCR), or its derivatives such as TaqMan assays [23]. Nucleic acid based approaches minimize false detection due to cross-reactivity issues [24] and are frequently coupled to isothermal pre-amplification of the viral RNA to increase sensitivity of detection [25] compared to serological methods. However, these methods increase the cost of testing and require trained personnel and specialized laboratory equipment, factors that render them unsuitable as methods for routine WNV detection in the field. 

The developing field of Synthetic RNA Biology recently contributed to the development of synthetic biosensors in the form of toehold switches [26], which can be utilized as novel ribodiagnostic tools. Inspired by natural riboswitches [27], toehold sensors consist of three domains: the 5’ sensing sequence that is complementary to an RNA target, followed by a ribosome binding site (RBS) loop in the middle and a downstream 3’ reporter sequence, an overall design that ensures increased stability and functionality compared to classical riboswitches [28]. In the absence of trigger RNA, the toehold folds into a stem-loop conformation due to a minimum number of internally designed complementary nucleotides. This closed conformation prevents ribosomal binding to the RBS and thus limits translational reporter output. Presence of the target RNA acts as a *cis*-acting trigger that unfolds the toehold switch due to hybridization at the 5’ sensing toehold sequence, a conformation that is thermodynamically preferred compared to the stem-loop folding of the resting toehold. Unfolded toehold enables ribosomal binding and subsequently translation of output signal that is proportional to the number of trigger RNAs [29]. Recently, toehold switches were successfully used for developing novel and cost effective, paper-based diagnostic tools for ZIKV, another flavivirus that is related to WNV [30]. 

Currently, no effective therapeutic treatment or vaccine is available against WNV [31,32]. Prevention of WNV infection relies solely on self-protective measures of citizens, complemented by routine yet expensive monitoring of WNV-infected vectors [21,33] and effective control of mosquito populations by the authorities [34]. Therefore, development of novel, cost-effective, sensitive and specific diagnostic methods, suited for widespread epidemiological surveillance of infected hosts and vectors, will contribute to early virus detection and prevention of future WNV outbreaks. Below we present an adaptation of the toehold technology for the detection of WNV. We used multiple phylogenetic, sequence and thermodynamic criteria to select thirty WNV toehold switches out of an original pool of 10,883 overlapping 35-mers across the WNV genome. We subjected our selected toehold switches to in vitro transcription/translation trials and selected the most efficient ones based on Green Fluorescence Protein (GFP) signal detection in the presence of isolated WNV genomic RNA. In parallel, we optimized an RT-qPCR-based detection and quantitation of WNV, with the aim of confirming, quantifying, and specializing viral detection also with laboratory-based methods. Our approach is the first step towards the development of a low-cost diagnostic kit, capable of routine and rapid detection of WNV-infected vectors, animal hosts or humans in the field, without the need of specialized equipment or trained personnel.

## 2. Materials and Methods

### 2.1. Development of a Toehold Switch Cloning Vector

The pGEM-3zf+ vector (Promega # P2271) was modified for the development of a versatile toehold cloning vector. Initially the T7 terminator sequence (TAGCATAA CCCCTTGGGGCCTCTAAACGGGTCTTGAGGGGTTTTTTGA) was cloned with PstI (Enzyquest, Heraklion, Greece #RE028S) and HindIII (Enzyquest, Heraklion, Greece #RE016S). Next, the enhanced GFP (EGFP) sequence (Addgene plasmid #176015) was amplified with proofreading Q5 polymerase (NEB M0491S) and subsequently cloned to the T7-term modified pGEM vector with HincII (NEB #R0103S). EGFP sequence was base-to-base confirmed with Sanger sequencing (CeMIA SA, Larissa, Greece) using the T7 promoter to provide the pGEM-T7-EGFP vector. All plasmid isolations were performed with the nucleospin plasmid mini kit (Macherey Nagel #740588.50) according to manufacturer protocol, or with homemade alkaline lysis protocols.

### 2.2. Bioinformatic Analysis and Selection of Appropriate WNV Sequences 

A collection of 617 WNV full genomic sequences were downloaded from Genbank (Appendix A), including 13 sequences that were reported in Greece (Appendix A). Fifteen additional full genomic sequences, corresponding to isolates of eight flaviruses that are related to WNV were also downloaded from Genbank (Appendix A). These sequences were aligned with Clustal Omega (including PIM calculation) and CLC Sequence Viewer (Qiagen) according to the following scheme: first the group of Greek WNV sequences was independently aligned to define hyper-conserved (100%) genomic regions indicative of regional conservation that also represents lineage 2 strains. Next, that consensus sequence was aligned against the group of the global WNV sequences to exclude variable genomic regions (<100% nucleotide identity). Subsequently, the same consensus sequence was aligned against the rest flavivirus sequences, with the aim of selecting sequence stretches with 100% nucleotide identity that significantly differentiate them from related viruses due to gaps or nucleotide mismatches, conferring less than 90% nucleotide identity. In parallel, the WNV consensus sequence was subjected to random k-mer fragmentation using R custom made code based on the CRAN package tcR (https://CRAN.R-project.org/package=tcR, accessed on 24 October 2018), resulting into a pool of 10,883 random and 1nt-overlapping 35-mers with 5’-3’ orientation. 35-mers that overlapped genomic regions with low WNV strain conservation or high *Flaviviridae* conservation were immediately eliminated from downstream selection. The remaining 35-mers were tested for Single Nucleotide Polymorphism (SNP) occurrence among the Greek WNV sequences, eliminating 35-mers with less than 100% conservation from the remaining lineage 2 strains, ensuring lineage 2 strain detection, including but not limited to the Greek strains. Next, the selected 35-mers (n = 194) were tested for conservation against each of the eight flaviviruses (accession numbers shown in Appendix A) or the human genome with Genbank BLAST alignment (blastn optimization for somewhat similar sequences), eliminating those with at least 1% sequence homology to another virus apart from WNV, providing the final filtered pool of 30 WNV-specific 35-mers. Construction of upset plots for summarizing 35-mer sequence identity results was performed with package upSetR [35] (https://github.com/hms-dbmi/UpSetR, accessed on 28 August 2021) in R.

### 2.3. Design and Cloning of the Toehold Switch Panel

The filtered pool of WNV 35-mers was converted to a pool of sense and antisense toehold oligos with NUPACK [36], after incorporation of RBS loop (ATAAAAGAGGAGAAATATGCT), downstream linker (TATAACCTGGCGGC AGCGCAAAAG) sequences and addition of 5’ EcoRI (Enzyquest, Heraklion, Greece #*RE014S)* and 3’ BamHI (Enzyquest, Heraklion, Greece #*RE005S)* restriction sites, facilitating downstream cloning. Toehold sequences were tested for thermodynamically unfavored structure, occurrence of stop codons or stretches of repetitive nucleotides with NUPACK, excluding unsuited oligos and providing the final pool of toehold sequences. Salt-free single stranded sense and antisense oligos from the selected final toehold pool were synthesised from Eurogentec, Liege, Belgium, renatured to 100 μΜ dsOligos using annealing buffer (10 mM Tris, 50 mM NaCl, 1mM EDTA) and ligated to the pGEM-T7-reporter vector with NEB T4 DNA ligase (M0202L) using supplied protocols. Proper cloning of the toehold sequence was confirmed with Sanger sequencing (CeMIA SA, Larissa, Greece). 

### 2.4. Cell Culturing of WNV and Viral RNA Extraction

All the procedures for WNV isolation were performed in the biosafety level 3 (BSL3) laboratory of the A’ Laboratory of Microbiology at the Medical school of the Aristotle University of Thessaloniki. Virus isolation was performed in Vero E6 cells; specifically, WNV stocks of the Greek strain 341/2010 (GenBank accession number KY594040) and strain Egypt 101 (GenBank accession number EU081844) diluted 1:10 in phosphate buffered saline (PBS) were added in 25-cm^2^ cell culture flasks covered by Vero E6 cell monolayers and the flasks were placed for one hour in an incubator at 37 °C with 5% CO_2_. The inoculum was then removed and 9 mL of culture medium consisting of minimum essential medium (MEM 10X, Gibco—REF: 21430-020) with 10% fetal bovine serum (FBS Qualified, Gibco, REF: 10270-098), L-glutamine 200mM (Gibco—REF: 25030-024), penicillin and streptomycin 5000 U/mL (Gibco—REF: 15070-063), amphotericin B 250μg/mL (Gibco—REF: 15290-018) and sodium bicarbonate solution 7.5% (Gibco—REF: 25080-060) was added. The presence of a cytopathic effect (CPE) was monitored daily. On day 6, when CPEs were present, a quantity of 140 μL of the supernatant (S/N) was used for RNA extraction, which was performed using the QIAamp Viral RNA kit (Qiagen, Hilden, Germany) according to manufacturer’s instructions. The presence of WNV was confirmed using the RealStar WNV RT-PCR Kit (altona Diagnostics GmbH, Hamburg, Germany). Viral titers are not available.

### 2.5. RNA Isolation from Blood or Plasma 

Total RNA was isolated from 500 μL blood or plasma of a healthy donor with TRI reagent (MRC, Cincinnati USA, #TR-118) according to manufacturer recommendations. 10 μg of total RNA were DNAse-treated with RQ DNAse I (Promega, Athens Greece #M6101) and subjected to phenol/chloroform isolation and ethanol precipitation. DNase treated RNA was spectrophotometrically quantified with the Quawell Q3000 spectrophotometer and electrophorized to confirm absence of genomic DNA prior to cDNA synthesis. 

### 2.6. cDNA Synthesis and Quantitative PCR

For qPCR standards, our designed amplicons of WNV lineage 1 (LIN1-, NC_009942.1) and lineage 2 (LIN2-, NC_001563.2) specific primer sets were amplified with KAPA HiFi Taq polymerase (KAPA BIOSYSTEMS #KK2101), following the same PCR protocol as for qPCR reactions (see below). Following gel extraction (MACHEREY-NAGEL #740609.50), the amplicons were phosphorylated with PNK (TAKARA, #2021A) for 30 min at 37 °C prior to phenol/chloroform and ethanol precipitation. 150 ng of the purified and phosphorylated LIN1 and LIN2 amplicons were cloned to 50 ng of EcoRV-digested and dephosphorylated (NEB rSAP #M0371S) pBluescript II KS+ vector (Stratagene, La Jolla, USA) using T4 DNA ligase (NEB # M0202L) and transformed to chemicompetent DH5α bacteria. Positive clones were selected with conventional PCR (KAPA BIOSYSTEMS #BK1002) using the LIN1 and LIN2 qPCR primers and verified with Sanger sequencing (CeMIA SA, Larissa, Greece). The confirmed LIN1 and LIN2 plasmids were used as 1:10 serially diluted spike-ins in 1000 ng DNAse-treated blood RNA constructing qPCR standard curves ranging from 1000 pg to 200 fg for LIN1 plasmid and 400 pg to 400 fg for LIN2 plasmid DNA respectively.

cDNA synthesis was performed with 100 ng of DNAse-treated total RNA from healthy donor plasma (virus free control) or with 100 ng of the same plasma RNA that was spiked with 100 ng LIN1 or LIN2 total RNA isolated from WNV (LIN1 or 2)-infected Vero E6 tissue culture supernatant. RT was performed with MMLV and primed with random hexamers according to manufacturer’s recommendation (Thermo Fisher Scientific, Athens, Greece #28025013). cDNA was diluted to 400 μl with ddH_2_O and 4 μl were used in triplicated qPCR reactions using KAPA SYBRGREEN qPCR mix (Sigma # KR0389_S). qPCR was performed in a Biorad CFX96 qPCR machine according to the following conditions: 95 °C −3 min, followed by 44 cycles of 95 °C −10 sec, 58 °C −20 sec, 72 °C −30 sec, followed by 72 °C −2 min and Melting Curve Analysis: 55.0 °C to 95 °C with increment of 0.5 °C for 0:05. Analysis was performed with the CFX software and/or Excel 365.

### 2.7. In Vitro-Transcription Translation

In vitro Transcription-Translation (IVT-TA) reactions were performed with PURExpress® kit (NEB #E6800L, Bioline Thessaloniki, Greece), as previously described [37], in the presence of 100ng WNV cultured S/N RNA unless otherwise stated in text. Signal detection was performed with Varioskan™ LUX multimode microplate reader (Thermo Scientific, Athens, Greece). Specifically, GFP excitation was performed at 488nm, while fluorescent emission was performed at 507, 509, 511 and 515 nm. Signal quantification was performed with the SkanIt RE 5.0 microplate reader software (Thermo Scientific, Athens, Greece). Statistical analysis and visualization were performed with R.

## 3. Results

### 3.1. K-mer Fragmentation of the WNV Full Genomic Sequence

Our k-mer strategy to identify appropriate trigger sequences for designing highly specific WNV toehold switches, resulted into a random fragmentation of the WNV reference genome sequence into short sequence stretches of 35 nucleotides, generating an initial pool of 10,883 overlapping 35-mers that were unbiasedly scattered across the WNV genome. Each 35-mer had the potential of serving as a toehold trigger, provided that additional criteria, such as sequence conservation within the WNV isolates, specificity for WNV compared to related flaviviruses and appropriate sequence composition and free-energy properties were fulfilled. 

### 3.2. Phylogenetic Analysis of WNV Full Genomic Sequence

Comparison of a series of WNV genomic sequences was performed with the aim of pinpointing ultra-conserved stretches of DNA sequences in the WNV genome that could be used as filtering criteria for selecting highly specific 35-mers from our original k-mer pool. To that purpose, a global collection of 603 WNV full genomic sequences (Appendix A), representing isolates from North and South America, North Africa and Middle East and Europe was subjected to phylogenetic analysis. Apart from geographical distribution, this collection includes isolates that were collected between 2000 and 2016 from 25 different species (both birds and mosquito vectors), corresponding to almost two decades of WNV evolution and multi-species distribution. These genomic sequences were aligned together with an available sub-collection of 14 full genomic sequences of Greek WNV isolates (Appendix A) to exclude DNA sequence stretches with mismatches in the Greek isolates, thus increasing specificity for the Greek WNV strains. 

As expected, the phylogenetic analysis clustered the Greek isolates together with the WNV lineage 2 reference sequence and other European isolates (Figure 1). A crude correlation between the phylogenetic clustering and the geographical origin of the WNV isolates was observed, with the Middle East clade clustering closer to the European clade and the Central American clade clustering closer to the North American isolates, also hosting the WNV lineage 1 reference sequence. We did not observe any significant correlation between strain clustering and vector species or year of isolation. Our phylogenetic approach allowed us to pinpoint an initial pool of 194 35-mers that are nested in hyper-conserved genomic stretches across the WNV genome, which can be used to design toeholds with increased specificity against a global pool of WNV isolates. Subsequently, we sought to further filter our 35-mers based on their overall similarity against the genome of eight flaviviruses that are related to WNV (Appendix A). Therefore, we expanded our phylogenetic approach to include the full genomic sequence of ZIKV, DENV, chikungunya, o’nyong-nyong, Semliki forest, Powassan (POWV), Saint Louis encephalitis (SLE) and yellow fever viruses, which we compared against the Greek WNV sequences and the reference WNV lineage 2 genomic sequence (Appendix A). Within the *Flaviviridae* family, DENV and ZIKV are more closely related to WNV, than the others, therefore we emphasized more on them by including two ZIKV and six DENV sequences for representing old and recently reported sequences from the typical serotypes of these viruses. Approximately 20% of our initial 194 pool, corresponding to forty 35-mers in conserved WNV DNA stretches, deviated significantly from the tested flaviviruses (Figure 2A), due to extensive nucleotide mismatches or sequence gaps within the 35-mer sequence (Appendix A). 

### 3.3. Sequence Analysis and Filtering of Toehold Switches

We subjected our filtered pool of forty 35-mers to sequence BLAST against the human genome and as expected we did not detect any significant sequence similarity. We then converted our filtered 35-mers to EGFP-reporter toehold switches and removed two toeholds due to premature stop codons. Next, we analyzed the sequence of our remaining pool of 38 toeholds in terms of GC content, melting temperature and nucleotide composition. This additional filtering step is important for detecting and excluding switches with atypical nucleotide or temperature melting properties. The average GC content of the toehold pool was equal to 49%, corresponding to an average Tm of 79 °C (Figure 2B). 

In addition, our toehold pool was balanced in terms of nucleotide composition with an average percentage of 22% cytosines, 27% guanines, 25% thymines and 26% adenines respectively (Figure 2C). Importantly, our selected switches correspond to 35-mers that span the whole genome of WNV, except for the NS2B and NS4A genes that are not covered by 35-mers due to sequence restrictions (Appendix A). This analysis revealed one toehold with significantly lower GC content of 20% and a melting temperature of 57.6 °C, which was excluded from downstream analysis. Interestingly, our toehold sequence properties are comparable to the published ZIKV toehold switches (GC = 46% and Tm = 78.5 °C), yet with a more balanced nucleotide composition (Appendix A). 

We also calculated the minimum free energy (MFE) of our selected thirty-eight toehold switches and compared them against our initial pool of 194 selected switches or against the twenty-four published ZIKV switches.

**Figure 2 genes-14-00237-f002:**
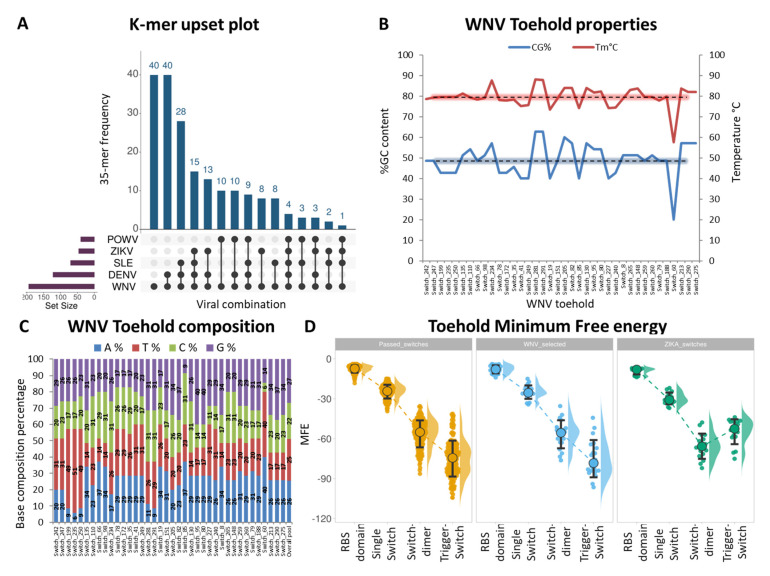
Design of WNV toehold switches (**A**) Upset plot highlighting the distribution of WNV 35-mer sequences across eight viruses that are related to WNV. 35-mer assignment was performed with BLAST and was scored as positive from as low as 1% of percent identity. Viruses with empty k-mer frequency are not shown. (**B**) Sequence properties of the filtered pool of 38 toeholds. Percentage of toehold sequence GC content is shown with the blue curve on the left y-axis. Melting temperature of the toehold sequence is shown with the red curve on the right axis. Dotted lines correspond to average values of the toehold pool. (**C**) Nucleotide composition of the filtered pool of toehold sequences. (**D**) Dot plot of Minimum Free Energy (MFE) calculations of four distinct toehold domains: the Ribosomal Binding Site (RBS), the single toehold switch, the toehold switch dimer and the Trigger-Switch dimer. Calculations include the total collection of 194 conserved k-mers (selected switches, shown with yellow), compared against the final pool of 38 selected WNV toeholds (blue) and the pool of 24 published ZIKV toeholds (green).

We focused MFE calculations for each toehold on the following four critical structures: the RBS domain that needs to be exposed in the presence of trigger RNA, the single switch that also needs to remain in a looped conformation in the absence of trigger, the switch dimer, formation of which reduces the available switch pool and therefore overall sensitivity of detection and finally the trigger-switch dimer that determines the conversion of transcriptional input to translational output. Based on MFE calculation, the most thermodynamically preferred structure on average was the trigger-switch, followed by the switch dimer and the single switch conformations (Figure 2D). As expected, the RBS domain showed the lowest stability due to sequence length differences compared to the remaining structures of the analysis. We observed a slight enrichment for lower MFE values in our selected WNV toeholds compared to the original pool, suggesting that our selection slightly enriched for more stable toehold-trigger interactions. Interestingly, we did observe a significant decrease on average trigger-switch MFE between our selected toeholds and the published ZIKV toeholds (Figure 2D). Encouraged by these results, we excluded eight switches with MFE trigger-switch values that exceeded pool average and continued with a final pool of 30 toehold switches for subsequent in vitro trials.

### 3.4. Detection of WNV with GFP Toehold Riboswitches

After selecting the final set of switch sequences, we first cloned them into GFP reporter vectors (see Materials and Methods) in order to construct the toehold switches and, subsequently, we performed IVT-TA trials using each one of the 30 toehold switches. We tested each toehold switch independently using 100 ng of RNA isolated from WNV LIN2 tissue culture supernatant as trigger (Figure 3A). We measured GFP fluorescence in four channels to challenge detection efficiency across the GFP emission spectrum. Overall detection was very stable regardless of emission spectrum, with channel 507 showing the lowest signal strength compared to the remaining channels. In terms of individual toehold performance, out of the original pool of thirty switches, 20% corresponding to six toeholds (T41, T134, T1376, T1874, T3046 and T10284) showed increased GFP signal ranging on average between 2.8- (T41) and 23-fold (T10284) above control average across the different emission channels (Figure 3B). Toeholds 41 and 134, showing 2.9- and 3.0-fold of GFP signal in channels 509–514 nm, did not deviate significantly from control average in channel 507 nm, which was the weakest in terms of overall signal detection. A second pool of six toeholds (T850, T2610, T4174, T6293, T7771 and T7834) showed marginal increase ranging between 1.5- and 2-fold over control average across the different channels, while the remaining 28 toeholds did not deviate significantly from control average. 

Subsequently, we mixed the six most prominent toehold switches and tested them together in IVT-TA reactions using 100 ng of isolated RNA from newly cultured WNV lineages 1 and 2 separately as trigger. The toehold mix showed on average 5.1- and 5.3-fold enrichment of GFP signal over control, a performance that was slightly lower than the average of 6.5-fold based on individual toehold trials across all channels (Figure 3C). In terms of lineage detection, our toehold mix did not discriminate between the two WNV lineage triggers. We individually tested T10284 that outperformed all other toeholds in the original individual toehold screen, with the newly cultured WNV lineage 1 and 2 triggers. Again, T10284 significantly outperformed the six-toehold mix, showing 12.4- and 11-fold enrichment over control. However, this fold enrichment significantly deviates for the performance of the 22-fold from the same toehold in the original screen of the individual switches (Figure 3A). Finally, similar to the toehold mix trial, we did not observe any significant discrimination between the two WNV lineage triggers in the individual performance of T10284.

**Figure 3 genes-14-00237-f003:**
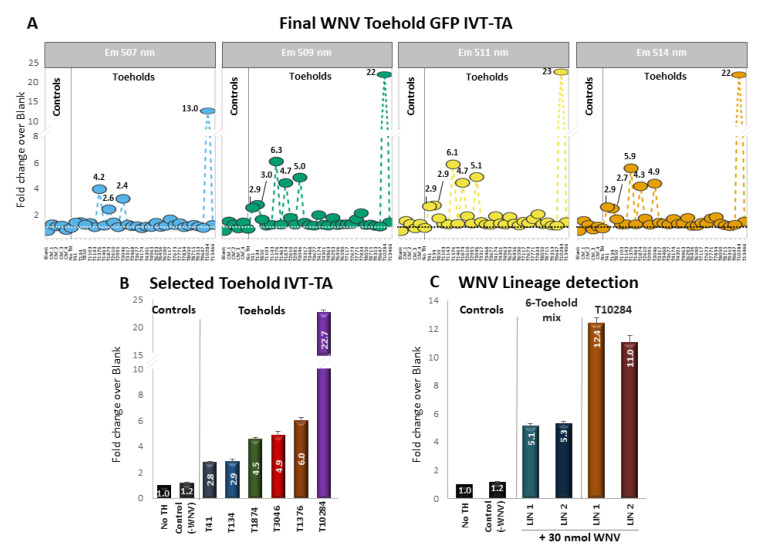
In vitro trials of WNV toehold switches (**A**) Individual in vitro Transcription/Translation (IVT-TA) trials of thirty selected GFP-toeholds of WNV. Signal detection was performed in four different channels, corresponding to 507 (blue), 509 (green), 511 (yellow) and 514 (orange) nm across the GFP emission spectrum. Y-axis shows fold enrichment of GFP detection compared to the average of negative controls. Negative controls include IVT-TA trials of different toeholds mixes in the absence of trigger RNA and IVT-TA without any WNV toehold in the presence of 100 ng WNV RNA. (**B**) Independent biological repetitions of the selected toeholds using 100 ng of WNV lineage 2 genomic RNA trigger (**C**) IVT-TA trials of mixed WNV toeholds using both lineage 1 and 2 isolated RNA as trigger. In addition, T10284 that outperformed all other toeholds was also tested individually against both lineage triggers.

### 3.5. Detection and Quantification of WNV RNA with qPCR

Parallel to IVT-TA reactions, we took advantage of our extensive phylogenetic analysis and our conserved 35-mer pool (Figure 1A and Appendix A) to design primers for detecting WNV with laboratory based molecular methods, such as quantitative PCR. We reckoned that qPCR-based detection would be essential as a sensitive method for confirming and quantifying WNV presence with laboratory standards after an initial toehold-based screen. To that purpose we designed two pairs of qPCR primers, specialized to detect either WNV LIN1 or LIN2 presence. 

We subjected both primer sets to absolute qPCR-based quantitation of WNV presence in healthy plasma RNA and confirmed that both LIN1 (Appendix A) and LIN2 (Appendix A) spike-ins were specifically detected with picogram to femtogram sensitivity. Using qPCR, we estimated that the 100 ng of tissue culture S/N RNA that we used as trigger in our toehold trials correspond to approximately 400 pg of WNV LIN2 RNA. Finally, the high specificity of both primer pairs for WNV detection was confirmed in qPCR reactions against ZIKV and DENV (Appendix A).

## 4. Discussion and Future Perspectives

Approximately 700 million humans are affected by mosquito-borne infectious diseases on an annual basis [38]. Consequently, 14% corresponding to 1 million infected patients die from geographically widespread flaviviruses, such as ZIKV, DENV and WNV, highlighting the importance of early viral detection for public health [39]. Although it was first isolated in Uganda in 1937 [1], for more than six decades it remained as an enzootic mosquito-borne disease in Africa, the Middle East, Russia and Europe, predominated by lineage 2 [40]. Epizootic outbreaks occurred in the mid 1990’s in Romania, other parts of Europe, Russia and Israel due to a new WNV lineage 1, which resulted in high proportion of neurological infections [40,41,42]. However, WNV rose to prominence after its arrival in New York city in 1999 and its spread throughout North America (for a review see [43]) and Canada, as well as elsewhere in Central and South America in less than ten years [17,44,45,46,47,48,49,50]. Ever since, it accelerates its global geographic distribution with increasing rate due to climate change-mediated expansion of its natural vectors [51]. Greece has been one of the most affected by WNV country in Europe. The disease was first detected in 2010 [35,36] with nearly 200 neuroinvasive cases associated with a high 17% mortality rate [37]. While cases fluctuated in the following years showing a diminishing trend towards 2015–2016 [15,38], a considerable rise occurred in 2018 throughout Europe with Greece and Italy accounting for 20% and 39% of the total number of reported cases, respectively [39]. In 2022, Italy and Greece kept the negative record of the first two most affected European countries (ECDC 2022). In this work we are combining a phylogenetic approach with synthetic riboswitch design to develop a pool of toehold riboswitches that are optimized for detection of WNV genomic RNA. 

Our phylogenetic analysis confirmed the geographical separation of WNV lineages 1 and 2, clustering the Greek strains into the lineage 2 clade, together with strains from Balkans (e.g Bulgaria, Figure 1). Previous reports classify WNV into up to nine lineages [52], with an average rate of nucleotide substitution per year, estimated at 7.55 × 10^−4^, highlighting the adaptive nature and expanding dynamics of the virus [53]. The prevailing lineages 1 and 2 have been associated with disease and significant epidemics in humans as mentioned above [54,55,56]. More specifically, lineage 1 strains demonstrate dispersed epidemiology, since they have been isolated from Europe, the Middle East, Africa, America and Australia [57,58]. Lineage 2 strains were originally limited in Africa and associated with asymptomatic infections [59]. Currently their geographical distribution is wider, extending to Europe where it causes large outbreaks at annual basis (ECDC 2022, [60,61,62,63]), with most cases being reported in Italy, Greece and Serbia. 

Nucleic-acid hybridization sensor approaches that are designed against viruses include the catalyzed hairpin assembly (CHA) and the hybridization chain reaction (HCR) [64]. Although both CHA and HCR were originally designed to detect miRNAs, they have been tested against viral cDNAs such as HIV [65] and Ebola [66]. Lately, riboswitches have been utilized in molecular ribodiagnostic applications [67]. Their RNA-centered design and regulation ensure flexibility and accuracy coupled with rapid and sensitive target detection [29,68]. More specifically, cell-free toehold approaches, combined with loop-mediated amplification and paper-based systems have been used to successfully detect and discriminate between the coronaviruses MERS-CoV and SARS-CoV-2 [69], responsible for the Respiratory Syndrome outbreaks in Middle East and South Asia respectively. In addition, toehold riboswitches fused to a colorimetric reaction were also successful in detecting and differentiating between the two subgroups of the Respiratory Syncytial virus, providing portable detection without the need for laboratory equipment [70]. Moving away from respiratory viruses, toehold-based detection was again used to detect Noroviruses, which are the primary cause of gastroenteritis, in clinical samples [71], or even plant pathogens such as the Potato virus Y [72]. Collectively, these examples highlight the effectiveness of toehold switches in viral ribodiagnostics. 

In our pilot attempt to design toehold switches against WNV, we focused on the relatively short reporter sequence of GFP. A shorter reporter sequence adds the benefit of increased protein production and therefore signal compared to longer reporter genes such as lacZ [30] that are commonly tested. This, however, limits the detection to laboratory equipment compared to classical colorimetric outputs. We have already begun experimenting with toehold designs that rely on colorimetric output from reporter genes with equally short sequences that are comparable to GFP. Furthermore, we complemented our toehold approach with the development of a sensitive molecular diagnostic method that can confirm WNV infection and quantify the viral load in selected samples from infected vectors or hosts under laboratory conditions. This stemmed from the need to accurately quantify WNV in a variety of tissues (from mosquito salivary glands, legs or ovaries to human blood, CSF or urine) in which viral load differs dramatically. We utilized our phylogenetic results as a guide for designing two primer pairs, capable of discriminating between lineage 1 and 2 strains with picogram sensitivity. Our preliminary qPCR results suggest specific detection of both WNV lineages at levels comparable to the reported clinical levels of the virus in circulating fluids of infected individuals. Collectively, we developed a whole-rounded strategy for detecting WNV consisting of a toehold-based synthetic biology approach for low-cost detection in the field, followed by a second qPCR-based approach capable of discriminating and quantifying the most predominant lineages of WNV. 

Our future plans include (i) optimization of the detection sensitivity of our toehold screens using decreasing amounts of synthetic WNV RNA as trigger in in vitro T-TA trials, (ii) expansion of the toehold-based trials to include isolated RNA from WNV-infected mosquitoes with the aim of challenging detection performance under more realistic conditions (iii) development streamlined laboratory approaches (e.g one-step qPCR protocol [73]) for rapid detection of WNV in a lineage-dependent manner and (iv) replacement of the existing GFP sequence with other reporter genes with the aim of converting the current fluorescent-based output to colorimetric reaction or even voltage generation that could be compatible with field detection without the need for laboratory equipment or specialized personnel. In addition, even though we computationally confirmed the specificity of our selected triggers against the related and cross-reacting in serology Usutu virus that is also abundant in Europe [74], we will expand our efforts to experimentally validate these computational predictions. Collectively these optimizations will facilitate the development of a low-cost method for detecting WNV in the field without the need of specialized laboratory equipment. 

## 5. Patents

A Secret-Know-How has been submitted to the Hellenic Copyright Organization (file registration number 2890, 02/02/2022 16:12:57).

## Figures and Tables

**Figure 1 genes-14-00237-f001:**
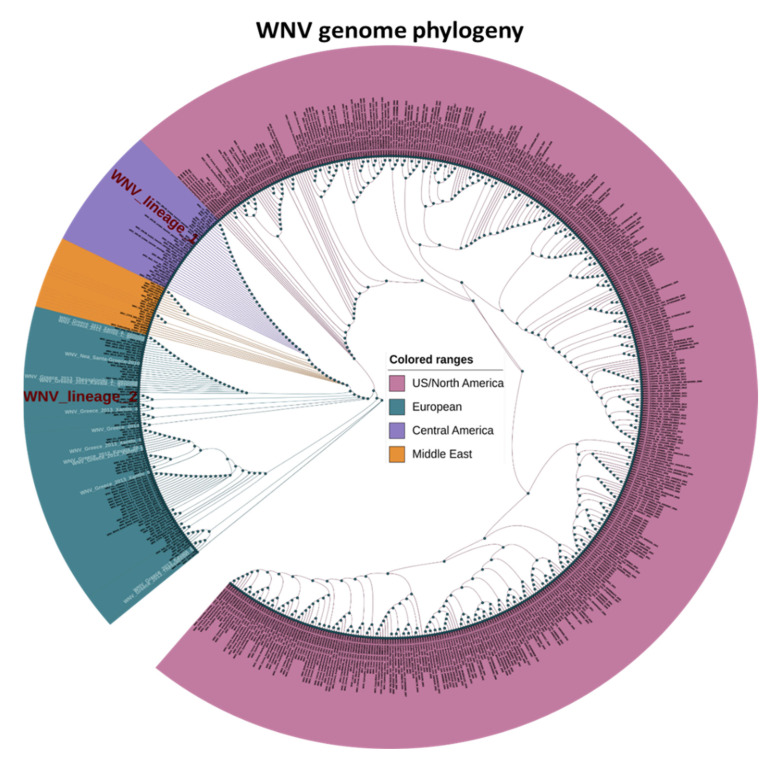
Phylogenetic analysis of WNV Maximum-likelihood phylogenetic tree analysis of 616 genomic sequences from a global collection of WNV isolated strains. Thirteen sequences from Greece are highlighted in white. All strains are coloured according to collection origin. The reference strains of WNV lineage 1 and 2 are also shown in red.

## Data Availability

All GenBank accession numbers are available in Appendix A.

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
