# Peer review of "Development of Toehold Switches as a Novel Ribodiagnostic Method for West Nile Virus"

_genes, 2023, doi:10.3390/genes14010237_

Round 1

Reviewer 1 Report

Recommendation - Accept with no revision

Author Response

We thank the reviewer for his/her very positive recommendation, which greatly encourages our efforts.

Reviewer 2 Report

Dear Author/s,

Submitted manuscript genes-2141840-v1 is well written draft but I want to draw the attention on the following area.

1.     Introduction is too lengthy and irrelevant up to some extent, epidemiological data can be shortened.

2.     In method section 2.1 It would be good if you can submit the plasmid sequence of vector and other to NCBI for better applicability or visibility for the researcher.

3.     Section 2.2 and others as well; need some references for detailing of methodology, if someone needed to look in to deeper. For example, k-mer fragmentation, cell culturing of WNV etc.

4.     Section 2.4 need to mention that blood/serum from infected patients or whatever.

5.     Section 2.5 LIN1 and LIN2 is a lineage number which belongs to section 2.2 or gene name?

6.     Section 2.6 IVT-TA reactions, abbreviation first described in section 3.6. I would suggest take care of all other abbreviations and minor mistakes like this.

7.     For the better visibility in figure 1, you can split it in to two figures viz. phylogenetic and other bioinformatics analysis.

8.     Figure 1B, mention POWV and SLE

9.     Discussion of the paper should be more prosperous and mention about the success of this technique in other trials or cases.

1.  Authors should pay more attention to make the manuscript more informative and crispier too. Take care of language and grammar at your end.

All the best.

Author Response

We thank the reviewer for his/her comments that significantly improved our revised manuscript. With regards to the raised comments :

  1. Introduction is too lengthy and irrelevant up to some extent, epidemiological data can be shortened

Author response : Well taken, we have shortened our revised introductory text, moving and modifying the phylogenetic and epidemiological description into the discussion (revised manuscript lines 497-537).

  1. In method section 2.1 It would be good if you can submit the plasmid sequence of vector and other to NCBI for better applicability or visibility for the researcher

Author response : Submission of the plasmid sequence of vector and other to NCBI, was not possible given the time constrains for the resubmission and due to restrictions of our secret know-how. However, the reviewer is right about the visibility, we therefore included the used T7 terminator sequence (line 149) and the reference for the used EGFP sequence (line 151) in section 2.1 of the revised M&M, hoping that the reviewer will appreciate our aforementioned constrains and consider these details together with the provided restriction enzymes that were used for their cloning as sufficient information at this stage.

  1. Section 2.2 and others as well; need some references for detailing of methodology, if someone needed to look in to deeper. For example, k-mer fragmentation, cell culturing of WNV

Author response : Well taken, we have added more details and a reference regarding the our k-mer fragmentation strategy (revised lines 177-178). With regards to cell culture, we think that the existing description of the WNV cell culturing (revised M&M, lines 210-228) part is quite detailed but if the reviewer has some specific suggestions for this section, we will be happy to include them if possible.

  1. Section 2.4 need to mention that blood/serum from infected patients or whatever

Author response : We clarified (in revised lines 230 and 258) that the blood/plasma samples are from a healthy donor, serving as a virus-free RNA for spike-in purposes.

  1. Section 2.5 LIN1 and LIN2 is a lineage number which belongs to section 2.2 or gene name?

Author response : We also clarified the usage of the LIN1 and 2 abbreviations in lines 240-241

  1. Section 2.6 IVT-TA reactions, abbreviation first described in section 3.6. I would suggest take care of all other abbreviations and minor mistakes like this.

Author response : We thank the reviewer for spotting this, we have cross-checked and corrected abbreviations such as IVT-TA throughout the text.

  1. For the better visibility in figure 1, you can split it in to two figures viz. phylogenetic and other bioinformatics analysis.

Author response : We modified Figure 1 according to the reviewer suggestion and indeed now the phylogeny is better presented to the reader and the context is better separated.  

  1. Figure 1B, mention POWV and SLE

Author response : We introduced both abbreviations in line 347.

  1. Discussion of the paper should be more prosperous and mention about the success of this technique in other trials or cases.

Author response : We have expanded our discussion session, providing several examples of nucleic-acid hybridization-based approaches (lines 538-542) or toehold applications in virus detection (lines 545-556).

  1. Authors should pay more attention to make the manuscript more informative and crispier too. Take care of language and grammar at your end.

Author response : We thank the reviewer for his/her spot-on comments and we believe that the revised manuscript better communes our research with the reader.

Reviewer 3 Report

Flaviviruses and their emergence are pressing the scientific community the development of novel, available and more-specific diagnostic methods. Ribodiagnostic brought amazing progress to the field and I am very grateful for the author’s decision to focus their interest on West Nile virus diagnostics. The methodology is very extensive and I appreciate the time and effort of the authors. I believe, the manuscript will be very interesting for the audience. However, the manuscript needs minor revision at this stage.

I see two limitations in the study design:

1.      The design of the method is suited predominantly to Greek WNV strains. Greece is one of the most affected countries in Europe and Greek strains are clustering also with e.g. Austrian, it will be much more reliable for e.g. Europe audience if you will focus on lineage 2.

2.      Within the flaviviral selection, you did not include Usutu virus. Even the Usutu virus is not expected to have such severe outcomes as WNV or DENV, you should consider its high abundance in Europe (e.g. asymptomatic blood donors) and high cross-reactivity in serology.

In the main text, please make sure any abbreviation is explained for the first use in the text. Also, please, add details about manufacturers, cities and countries for all used reagents, kits, and equipment.

Line 81 – 96 – very valuable data, maybe for some review about WNV in Greece? Here, it makes sounds like your manuscript is locally oriented

Line 105 –  add USUV and TBEV

Line 167 –  please change NCIB to GenBank. Genbank belongs to NCIB

Line 171 – make clear if all used sequences are available in Genbank

Line 186 – abbreviation needs to be explained somewhere, SNP

Line 188 – (n=194)

Line 189 – Accession numbers

Line 194 – add to reference list

Line 209 – company name, city, country

Line 210 – please add titers of your new WNV isolates

Line 240 – specify origin of WNV LIN1 and LIN2 amplicons

Line 473 – 486 – Regards to qPCR, I do not believe this study brought any development for WNV qPCR.

Firstly, the use of WNV standards is nothing new for the audience. e.g. https://www.frontiersin.org/articles/10.3389/fcimb.2020.619071/full

Secondly, described qPCR is quite “old fashioned”.  The authors performed 2-steps reverse transcription and separate RNA extraction. As was showed during COVID-19 pandemic and also is an ongoing trend in diagnostics, commercially available kits which allow the use of direct RNA and one-step RT-qPCR are very popular due to decreased time, capacities and also costs. https://www.nzytech.com/products-services/molecular-diagnostics/human-pathogen/vector-borne-diseases/md03971/

https://www.frontiersin.org/articles/10.3389/fmicb.2016.00554/full

Limitation of the staty – designed dominantly for Greek WNV, exclution of USUV and TBEV during the specificity design

Author Response

We thank the reviewer for his/her comments that significantly improve the revised manuscript and for appreciating our WNV ribodiagnostic efforts, which is very encouraging for us. With regards to the mentioned limitations of the study :

  1. The design of the method is suited predominantly to Greek WNV strains. Greece is one of the most affected countries in Europe and Greek strains are clustering also with e.g. Austrian, it will be much more reliable for e.g. Europe audience if you will focus on lineage 2.

Author response : Well taken, however we would like to kindly explain that our experimental design simply excluded 35-mers with SNPs among the Greek strains. Therefore, the selected 35-mers are suited for lineage 2 strain detection, including but not limited to the Greek strains. We have revised our 2.2 section in the M&M accordingly (lines 181-184), hoping that it is now clearer.

  1. Within the flaviviral selection, you did not include Usutu virus. Even the Usutu virus is not expected to have such severe outcomes as WNV or DENV, you should consider its high abundance in Europe (e.g. asymptomatic blood donors) and high cross-reactivity in serology.

Author response : With regards to the exclusion of USUV, the reviewer is most-certainly right. Given the short window for resubmission, we were unable to repeat the whole analysis with the inclusion of USUV but we cross-checked our final triggers against the USUV genome ensuring their specificity at least computationally. For now, we have added the experimental confirmation of this result as reference point for our future plans (revised lines 589-592).

Regarding the remaining points that were raised by the reviewer :

  1. In the main text, please make sure any abbreviation is explained for the first use in the text. Also, please, add details about manufacturers, cities and countries for all used reagents, kits, and equipment.

Author response : We cross-checked the manuscript for proper use of abbreviations and added the requested information in reagents and kits, wherever that was feasible.

  1. Line 81 – 96 – very valuable data, maybe for some review about WNV in Greece? Here, it makes sounds like your manuscript is locally oriented

Author response : We thank the reviewer for appreciating these data. We decided to move a shorter version of the reference of the WNV epidemiology in Greece (original lines 81-96) to the discussion (revised lines 508-515) instead of excluding it. Our intention is certainly not to orient the manuscript towards a Greek audience but to emphasize its purpose given the severity of WNV spread in this country (2nd most affected in Europe for 2022) and in the Balkan area in general.

  1. Line 105 – add USUV and TBEV

Author response : We added a reference for USUV and TBEV in revised line 94.

  1. Line 167 – please change NCIB to GenBank. Genbank belongs to NCIB

Author response : We changed all NCBI references to GenBank (revised lines 161 and 187).

  1. Line 171 – make clear if all used sequences are available in Genbank

Author response : We also clarified in the text (revised line 164) that all supplementary table 3 accessions are from GenBank (also stated inside the table, with accession codes)

  1. Line 186 – abbreviation needs to be explained somewhere, SNP

Author response : Included in revised line 181.

  1. Line 188 – (n=194)

Author response : Corrected in revised line 185.

  1. Line 189 – Accession numbers

Author response : Clarified in revised line 186.

  1. Line 194 – add to reference list

Author response : Added reference in revised line 191.

  1. Line 209 – company name, city, country

Author response : Edited in revised lines 155, 207 and 252.

  1. Line 210 – please add titers of your new WNV isolates

Author response : Unfortunately, we do not have a calculation of our viral titers (which is now stated in revised line 228), but as can be observed in Sup. Fig 2 A and C, the 100 ng of WNV S/N (that served as spike-ins) are detected in 20 and 17 Cts respectively. 

  1. Line 240 – specify origin of WNV LIN1 and LIN2 amplicons

Author response : Clarified in revised lines 240-241.    

  1. Line 473 – 486 – Regards to qPCR, I do not believe this study brought any development for WNV qPCR.

Author response : Our intention in this manuscript is to focus on WNV toehold development as a laboratory-free monitoring approach in the near future, not to develop a novel PCR-based diagnostic method. Nevertheless, parallel to our toehold optimization, we present our RT-qPCR results as a laboratory-based approach that can be optimized, to commune that our intention is to end up with a whole-rounded strategy for field monitoring of WNV along with the laboratory validation for both of its lineages. We have revised our discussion (lines 583-585) to make clear that our efforts continue also towards that direction.